# Src-Dependent NM2A Tyrosine Phosphorylation Regulates Actomyosin Remodeling

**DOI:** 10.3390/cells12141871

**Published:** 2023-07-17

**Authors:** Cláudia Brito, Joana M. Pereira, Francisco S. Mesquita, Didier Cabanes, Sandra Sousa

**Affiliations:** 1i3S-Instituto de Investigação e Inovação em Saúde, Universidade do Porto, 4200-135 Porto, Portugaljmpereira@ibmc.up.pt (J.M.P.);; 2IBMC, Instituto de Biologia Celular e Molecular, 4200-135 Porto, Portugal; 3MCBiology PhD Program–Instituto de Ciências Biomédicas Abel Salazar-ICBAS, University of Porto, 4050-313 Porto, Portugal

**Keywords:** NM2A, Src kinase, actomyosin cytoskeleton, cytoskeletal remodeling, cell migration

## Abstract

Non-muscle myosin 2A (NM2A) is a key cytoskeletal enzyme that, along with actin, assembles into actomyosin filaments inside cells. NM2A is fundamental for cell adhesion and motility, playing important functions in different stages of development and during the progression of viral and bacterial infections. Phosphorylation events regulate the activity and the cellular localization of NM2A. We previously identified the tyrosine phosphorylation of residue 158 (pTyr^158^) in the motor domain of the NM2A heavy chain. This phosphorylation can be promoted by *Listeria monocytogenes* infection of epithelial cells and is dependent on Src kinase; however, its molecular role is unknown. Here, we show that the status of pTyr^158^ defines cytoskeletal organization, affects the assembly/disassembly of focal adhesions, and interferes with cell migration. Cells overexpressing a non-phosphorylatable NM2A variant or expressing reduced levels of Src kinase display increased stress fibers and larger focal adhesions, suggesting an altered contraction status consistent with the increased NM2A activity that we also observed. We propose NM2A pTyr^158^ as a novel layer of regulation of actomyosin cytoskeleton organization.

## 1. Introduction

The non-muscle myosin 2A (NM2A) is a major cytoskeletal motor protein that plays a central role in the contractility of the actin cytoskeleton [1,2]. Each NM2A molecule is a hexameric complex composed of two non-muscle myosin heavy chains (NMHC2A) and two pairs of light chains: the regulatory (RLCs) and essential (ELCs) light chains [1,2]. NMHC2A folds into three different domains: the conserved head/motor domain comprising the sites for ATP hydrolysis and actin binding; the neck domain that interacts with the light chains, stabilizing the protein and regulating NM2A activity; and the tail, containing a coiled coil region responsible for heavy chain homodimerization and non-muscle myosin II (NMII) filament formation [2,3]. Phosphorylation of the RLCs at threonine 18 and serine 19 regulates the NM2A enzymatic activity and filament formation [4,5,6,7]. The inactive form of NM2A, associated with unphosphorylated RLCs, folds into a compact structure in which motor heads and tails directly interact [2,8,9]. This autoinhibited molecule does not form filaments, interacts only weakly with actin, and freely diffuses in cells [10,11,12,13]. Upon phosphorylation of the RLC, the active NM2A molecules undergo antiparallel self-association to form bipolar filaments [10,14], which interact with actin filaments and power actomyosin network contraction by converting chemical energy into mechanical force via ATP hydrolysis [15]. NM2A subcellular distribution and filament assembly are additionally regulated by the specific phosphorylation on RLC tyrosine 155 in migrating cells [16] and on serine and threonine residues at the NMHC2A tail domain [17,18].

NM2A is broadly required for processes dependent on force generation and remodeling of the actomyosin cytoskeleton [19,20]. NM2A self-organizes into ordered superstructures, such as super clusters and stress fibers, through a combination of mechanical interactions and biochemical signaling [21,22,23,24]. Its activity and self-organization properties modulate the architecture of actomyosin networks, thus determining its function. In addition to regulating NM2A activity, phosphorylation plays critical roles in its organization within actomyosin networks [25,26]. The remodeling and contractility of such actomyosin networks shape cells and tissues during physiological processes such as development and wound repair.

Defects in NM2A functions are associated with a multitude of human diseases [22,27,28], including various viral and bacterial infections [29,30,31]. Upon cellular infection by the human pathogen *Listeria monocytogenes*, Src kinase is activated [32] and NMHC2A is phosphorylated in the conserved tyrosine 158 (Tyr^158^) residue, located in its motor domain [30]. Phosphorylation of Tyr^158^ (pTyr^158^) occurs in a Src-dependent manner and restricts bacterial cellular invasion [30]. Whether this phosphorylation controls the cellular processes NM2A are involved in, thus regulating the active cellular remodeling of actomyosin networks in space and time, has not been addressed yet.

Herein, we investigated the impact of NMHC2A pTyr^158^ on actomyosin cytoskeleton organization, cell adhesion, and migration. We found that the Src-dependent regulation of pTyr^158^ is essential to control the organization of the actomyosin cytoskeleton, assembly/disassembly of focal adhesions, and cell motility. Our results provide new insights into NM2A regulation through a novel phosphorylation event, which modulates subcellular events where NM2A’s function is crucial.

## 2. Materials and Methods

### 2.1. Cell Lines

HeLa cells (ATCC CCL-2) were grown in DMEM with glucose and L-glutamine (Lonza), supplemented with 10% fetal bovine serum (FBS; Biowest, Nuaillé, France), and maintained at 37 °C in a 5% CO_2_ atmosphere.

### 2.2. Reagents, Toxins, Antibodies and Dyes

Dasatinib (Santa Cruz Biotechnology, Dallas, TX, USA) was used at 300 nM in complete medium for 1 h before the experiment. The following antibodies were used at 1/200 for immunofluorescence microscopy (IF) or 1/1000 for immunoblotting (IB): mouse anti-β-actin (#A1978, Sigma, St Louis, MO, USA); rabbit anti-NMHC2A (#M8064, Sigma, St Louis, MO, USA); rabbit anti-calnexin (#AB2301, Merck Millipore, Burlington, MA, USA), rabbit anti-vinculin (#700062, Fisher Scientific, Waltham, MA, USA); rabbit anti-pMLC (Thr18/Ser19) (#3674, Cell Signaling, Danvers, MA, USA); and monoclonal anti-Src (GD11, abcam, Cambridge, UK). For IF analysis, DNA was stained with 4′,6-diamidino-2-phenylindole dihydrochloride (DAPI; Sigma, St Louis, MO, USA) and actin with Rhodamine Phalloidin (Thermo Fisher Scientific, Waltham, MA, USA) at 1/200. Secondary antibodies were used at 1/500: goat anti-rabbit Alexa Fluor 488 (Invitrogen, Waltham, MA, USA), goat anti-rabbit Alexa Fluor 594 (Invitrogen, Waltham, MA, USA), and goat anti-mouse Cy3 (Jackson ImmunoResearch, West Grove, PA, USA). For IB, goat anti-rabbit or anti-mouse HRP (Abliance, Compiègne, France) was used at 1/10,000.

### 2.3. Plasmids

CMV-GFP-NMHCII-A plasmid was a gift from Robert Adelstein (Addgene # 11347) [33]. Insertion of point mutations Y158F and -Y158E was achieved by site-directed mutagenesis using a QuickChange II Site-directed mutagenesis kit (Agilent, Santa Clara, CA, USA) as described in [30]. Hsp90 HA was a gift from William Sessa (Addgene # 22487) [34].

### 2.4. Transfections and shRNA Lentiviral Transductions

HeLa cells seeded on top of glass coverslips in Nunc^TM^ 24-well plates (5 × 10^4^ cells/well), into Ibitreat μ-dishes (Ibidi, Martinsried, Bayern, Germany) (1 × 10^5^ cells/well), or Nunc^TM^ 6-well plates (5 × 10^5^ cells/well) were, respectively, transfected with 0.5 μg, 1 μg, or 2.5 μg of DNA using jetPRIME^®^ Polyplus transfection reagent or Lipofectamine 2000^®^ (Thermo Fisher Scientific, Waltham, MA, USA) according to the manufacturers’ instructions. Protein expression was allowed for 18 to 24 h before cells were processed. shRNA lentiviral transductions were performed as described in [30] and the sequences used are available in the Appendix A.

### 2.5. Immunoblot Assays

HeLa cells total protein extracts were recovered in sample buffer (0.25 mM Tris–Cl, pH 6.8; 10% SDS; 50% glycerol; and 5% β-mercaptoethanol), resolved by SDS–PAGE (10% acrylamide), and transferred onto nitrocellulose membranes using a TransBlot Turbo™ system (BioRad, Hercules, CA, USA) at 0.3 A for 1 h. Primary and secondary HRP-conjugated antibodies were diluted in TBS–Tween 0.1% (150 mM NaCl; 50 mM Tris–HCl, pH 7.4; and 0.1% Tween) with 5% (*m*/*v*) milk. For the immunoblot assessing the expression levels of Src, 20 ug of total protein extracts from WT and shSrc HeLa cells were loaded. Washes were performed with TBS–Tween 0.2%. The signal was detected using Western Blotting Substrate (Thermo Fisher Scientific, Waltham, MA, USA) and collected in a ChemiDoc™ XRS+ System with Image Lab™ Software (version 3.0.1.14, BioRad, Hercules, CA, USA).

### 2.6. Immunofluorescence Microscopy

Cells were fixed in 3% paraformaldehyde (15 min at room temperature), quenched with 20 mM NH_4_Cl (1 h), and permeabilized with 0.1% Triton X-100 (5 min). Coverslips were incubated for 1 h with primary antibodies, washed three times in PBS 1×, and incubated for 45 min with secondary antibodies and, when indicated, Rhodamine Phalloidin and DAPI. Antibodies and dyes were diluted in PBS containing 1% BSA. Coverslips were mounted onto microscope slides with Aqua-Poly/Mount (Polysciences, Warrington, PA, USA). Images were collected with a confocal laser scanning microscope (Leica SP5 II, Wetzlar, Germany) and processed using Fiji™ (version 2.9.0/1.53t) or Adobe Photoshop (CS6 version 13.0) softwares.

### 2.7. g-Actin and f-Actin Separation

HeLa cells (5 × 10^5^) were seeded in six-well plates and either transfected to express GFP-NMHC2A-WT or -Y158F variants or treated with dasatinib for 1 h. Cells were washed twice with PBS 1x and homogenized for 10 min at 37 °C in actin stabilization buffer (0.1 M Hepes, pH 6.9, 30% glycerol (*v*/*v*), 5% DMSO (*v*/*v*), 1 mM MgSO_4_, 1 mM EGTA, 1% Triton X-100 (*v*/*v*), 1 mM ATP, complete protease, and phosphatase inhibitors). Total protein fractions were ultracentrifuged at 100,000× *g* for 75 min at 37 °C. Supernatants containing g-actin were collected, and pellets containing f-actin were resuspended in an equivalent volume of RIPA buffer. Total, supernatant, and pellet fractions were processed for immunoblotting.

### 2.8. Quantification of Immunofluorescence Images: NMHC2A Aggregates, Stress Fibers, Focal Adhesions, Phospho-Regulatory Light Chain (pMLC), Cell Area, and Aspect Ratio

The percentage of cells displaying stress fibers or NMHC2A aggregates was quantified in at least 200 cells per sample. Cells showing approximately 50% (arbitrary) of their total area covered by dense actomyosin fibers were considered positive for stress fibers. Cells displaying at least two distinct aggregates larger than 2 µm or one larger than 5 µm were considered positive for aggregation. Anisotropy was quantified using the Fiji™ plug-in FibrilTool as described in [35] and focal adhesions were quantified as described in [36] using the CLAHE and Log3D Fiji™ plug-ins. The levels of pMLC were quantified using Fiji™ by measuring the raw integrated density of pMLC in each cell normalized by the total cell area in at least 100 cells per sample. The area of the cells and respective aspect ratio were measured using the analysis tool in Fiji™.

### 2.9. Cell Motility Analysis

Cells were seeded into Ibitreat μ-dishes (Ibidi, Martinsried, Bayern, Germany), transfected to express the different GFP-NMHC2A variants, maintained in OptiMEM (Gibco, Thermo Fisher Scientific, Waltham, MA, USA) at 37 °C and 5% CO_2_, and imaged using a widefield microscope (Leica DMI6000 Time-lapse, Wetzlar, Germany) equipped with a 20 × 0.4 NA objective. Data sets of transmitted light and the fluorescence of GFP-NMHC2A were acquired every 10 min between 16 and 20 h. Fiji™ was used for image sequence analyses and video assembly. The velocity and accumulated distance were analyzed in at least 200 cells/condition using the Manual tracking plug-in from Fiji™.

### 2.10. Fluorescence Recovery after Photobleaching (FRAP)

HeLa cells were transfected to express either GFP-NMHC2A-WT or -Y158F. Images were acquired 20 h after transfection using a laser scanning confocal microscope SP8 (Leica) equipped with a 63 × 1.3 NA objective at 37 °C and 5% CO_2_. Five pre-bleach images were acquired every 138 ms, followed by five bleaching scans with 100% intensity on the 488 nm laser line over the region of interest (ROI). The recovery of fluorescence was monitored for 56 s every 138 ms (400 frames), followed by 60 s every 300 ms (200 frames). Mean fluorescence intensities (MFI) were measured in Fiji™ and a fluorescence recovery analysis was performed as described in [37] using the easyFRAP-web on-line platform (https://easyfrap.vmnet.upatras.gr/ (accessed on 3 June 2019)). Briefly, the MFI of the bleached ROI was normalized to non-bleached ROIs. We followed a full-scale normalization which subtracts background values at each time point and corrects for laser fluctuations, fluorescence losses during photobleaching, differences in starting intensities, and losses in total fluorescence. Additionally, the MFI was corrected for differences in the bleaching efficiency following easyFRAP-web on-line platform recommendations [37]. The mobile fraction and the half-maximal recovery time (T-half) values were obtained by curve fitting. Curve fitting was performed using a single- and double-term exponential equation and obtained R2 values served as a goodness-of-fit measure. Only cells for which fitted curves showed an R2 ≥ 0.8 were considered to quantify the mobile fraction and T-half.

### 2.11. Statistical Analysis

Statistical analyses were carried out with Prism 8 (version 8.1.1, GraphPad Software, La Jolla, CA, USA), using a two-tailed unpaired Student’s *t*-test for comparison of means between two samples.

## 3. Results and Discussion

### 3.1. Actomyosin Cytoskeleton Organization Is Modulated by NMHC2A pTyr^158^

Previous data demonstrated that *Listeria monocytogenes* infection activates Src kinase [32], which in turn phosphorylates NMHC2A on the conserved tyrosine 158 (Tyr^158^) residue [30]. Considering the central role of NM2A in key cellular processes [2], we hypothesized that NMHC2A pTyr^158^ may control physiological events that rely on the remodeling of the actomyosin cytoskeleton. To address this, we tested the impact of NMHC2A pTyr^158^ in the overall organization of the actomyosin cytoskeleton. HeLa cells were transfected to express either a GFP-tagged wild-type NMHC2A (GFP-NMHC2A-WT), a mutant mimicking a permanent phosphorylation in which Tyr^158^ was replaced by a glutamate (GFP-NMHC2A-Y158E), or a non-phosphorylatable NMHC2A mutant in which Tyr^158^ was substituted by a phenylalanine (GFP-NMHC2A-Y158F). The percentage of transfected cells (Figure 1A) and the expression levels, measured as the mean GFP signal intensity (Figure 1B), were similar for the three NMHC2A variants.

A confocal microscopy analysis of transfected cells revealed that GFP-NMHC2A-Y158E aggregated in 40% of the cells, while GFP-NMHC2A-WT was detected in aggregates in 20% of the cells (Figure 1C,D). This aggregation phenotype, of which an acute example is depicted in Figure 1C, was reverted when GFP-NMHC2A-Y158E was co-expressed with the chaperone HSP90 (Figure 1E), reported to facilitate NMHC2A folding and actomyosin filament assembly [38,39,40]. This observation suggests that the aggregation phenotype might be related to GFP-NMHC2A-Y158E folding and/or assembly defects, which are overcome by the overexpression of HSP90. NM2A aggregation was previously reported for mutants affecting ATPase’s activity and ability to translocate actin filaments [41]. Thus, considering that GFP-NMHC2A-Y158E aggregation may trigger toxic effects and/or be associated with a decreased NM2A activity and that phospho-mimetic mutations do not always mimic the presumed phosphorylations [42,43], we pursued our study with only the non-phosphorylatable mutant (GFP-NMHC2A-Y158F).

About 60% of the cells expressing GFP-NMHC2A-Y158F were found enriched in actin filaments, displaying large areas covered by stress fibers (Figure 1C,F) organized in an anisotropic orientation (Appendix A). Only 20% of the GFP-NMHC2A-WT-expressing cells were enriched in stress fibers, which also displayed anisotropic orientation (Figure 1C,F and Appendix A). To support these findings, we performed actin fractionation assays to assess the levels of globular (g-) and filamentous (f-) actin in cells overexpressing NMHC2A-WT or -Y158F variants. The amount of f-actin (pellet fraction, P) was higher in GFP-NMHC2A-Y158F- than in GFP-NMHC2A-WT-expressing cells (Figure 1G) and thus, the g-/f-actin ratio was significantly decreased in cells expressing the GFP-NMHC2A-Y158F variant (Figure 1H). Our data thus show that cells expressing the non-phosphorylatable NMHC2A variant accumulate stress fibers and f-actin filaments. Since the total levels of actin remained comparable (Figure 1G), the accumulation of stress fibers and f-actin detected in cells expressing the non-phosphorylatable NMHC2A variant (GFP-NMHC2A-Y158F) may be due to increased actin polymerization and/or decreased depolymerization, which would alter the filament turnover and dynamics. We speculate that GFP-NMHC2A-Y158F may have a higher f-actin crosslinking activity than its WT counterpart, inducing filaments to coalesce and thus increasing the overall density of the network and contraction [44]. Besides motor activity, the NMHC2A-mediated crosslinking of actin filaments was proposed to increase tension and to contribute to cortical integrity [44,45,46]. Additionally, cells with stress fibers have been demonstrated to produce high contractile forces which are not locally restricted but propagate to the surrounding actin network, impacting the overall contractile energy of the cell [47]. The accumulation of f-actin or stress fibers is associated with an increased cell stiffness and rigidity, adhesion to the substratum, and decreased cell migration capabilities [48,49]. Our data thus suggest that by accumulating stress fibers and reorganizing their actomyosin cytoskeleton, cells expressing the non-phosphorylatable variant GFP-NMHC2A-Y158F may display strong adhesion to the substratum and an impaired ability to migrate. In addition, it raises the hypothesis that through the phosphorylation of NMHC2A Tyr^158^, the cell may reorganize its actomyosin cytoskeleton and adapt its contractile state to respond to external cues.

### 3.2. Src Kinase Coordinates Actomyosin Cytoskeleton Remodeling

Given that the phosphorylation status of Tyr^158^ affects the organization of the actomyosin cytoskeleton and that during *Listeria* infection, NMHC2A pTyr^158^ depends on Src kinase [30], we further investigated the role of Src activity in the overall organization of the actomyosin cytoskeleton. For that, we interfered with the Src activity by using the pharmacological inhibitor Dasatinib (Dasa) or downregulating the Src expression through specific shRNAs (shSrc). The reduced expression of Src kinase in shSrc cells was confirmed by immunoblotting (Appendix A). A confocal microscopy analysis consistently showed that cells with a reduced Src activity or Src expression display more stress fibers than control cells (Figure 2A). Indeed, the percentage of cells displaying stress fibers significantly increased under impaired Src conditions (Figure 2B). In addition, actin fractionation assays on total lysates of Src-impaired cells showed decreased levels of g-actin (supernatant fraction, S) and, concomitantly, increased levels of f-actin in the pellet fraction (Figure 2C). In agreement with the increased stress fibers detected by fluorescence microscopy, the g-/f-actin ratio was significantly reduced in Src-impaired cells (Figure 2D). These data show that cells with impaired Src activity display an accumulation of stress fibers, as observed in those expressing the non-phosphorylatable GFP-NMHC2A-Y158F. Of note, our previous data demonstrated that NMHC2A pTyr^158^ triggered by bacterial infection does not occur in cells expressing reduced levels of Src kinase [30]. Supporting this crosstalk between Src and NMHC2A, previous studies showed that Src activity controls cortical tension and/or membrane dynamics in NMII-dependent processes such as non-apoptotic plasma membrane blebbing [50], cell volume regulation upon osmotic stress [51,52], and neuronal growth cone [53].

A high Src activity reduces cell adhesion by triggering stress fiber disassembly [54,55,56], which enhances cell migration, invasion, and tumor metastasis [48,57]. Contrarily, and in agreement with our observations, a low Src activity was reported to promote stress fiber assembly, which is associated with cell stiffening [48] and a reduced cell migration [58]. In line with this, Src-deficient fibroblasts show reinforced integrin and actomyosin cytoskeleton interactions [59], also limiting migration. Our data strikingly show that the overexpression of a NMHC2A molecule non-phosphorylatable on Tyr^158^ phenocopies the behavior of cells with impaired Src activity, which suggests that Src-dependent phosphorylation of NMHC2A Tyr^158^ might contribute to Src-mediated cell processes. Thus, this raises the hypothesis that, besides coordinating the actomyosin cytoskeleton by targeting ROCK and FAK [60], Src kinase might also regulate the actomyosin cytoskeleton through the direct control of NMHC2A pTyr.

### 3.3. The Focal Adhesion Morphology Is Regulated by NMHC2A pTyr^158^

Stress fibers and focal adhesions are structurally and functionally interdependent [49,61]. Despite controversy in the field, much evidence pinpoints that the assembly of stress fibers and focal adhesions depends on tension generated by non-muscle myosin II (NMII) [49]. Indeed, the activity of NMII was shown to drive the assembly of stress fibers and focal adhesions [62] and NMII expression is required for the maturation of nascent adhesions [63,64]. Considering the described interconnections between Src, NMII activity, stress fibers, and focal adhesions and our observations regarding the stress fiber accumulation in cells expressing the non-phosphorylatable GFP-NMHC2A-Y158F variant, we investigated whether focal adhesions would also be affected by the status of NMHC2A pTyr^158^. Cells overexpressing GFP-NMHC2A-WT or -Y158F were fixed and immunolabeled for vinculin, a marker for focal adhesions [65]. The number, size, and shape of the focal adhesions were analyzed by confocal microscopy. Although in GFP-NMHC2A-Y158F-expressing cells the number of focal adhesions per cell was significantly decreased (Figure 3A,B), they were larger and more elongated, displaying increased areas and aspect ratios (Figure 3A,C,D).

The traction force that focal adhesions exert on the cell substrate was shown to be directly related to their size [66,67]. Possibly, under increasing tension, new proteins are recruited to focal adhesions whose composition may thus be regulated by mechanical stress [68,69]. In agreement with this, several proteins such as vinculin were found to be associated with focal adhesions in an NMII-dependent manner [70,71,72].

While the assembly of ventral stress fibers and focal adhesions depends on the tension powered by NM2A contractility [62,63,64,70], other studies reported that the tension and NMII activity had little effect on both stress fiber and focal adhesion assembly [73,74,75]. Our data favor the involvement of NM2A-mediated mechanical tension in the assembly of stress fibers and their associated focal adhesions.

Together with the increased stress fibers detected in cells overexpressing GFP-NMHC2A-Y158F, our results also suggest that focal adhesions are under increased tension and that the NMHC2A pTyr^158^ status not only regulates stress fiber assembly but also interferes with focal adhesion morphology, thus indicating a role in their formation and/or maturation. Overall, we show here the critical importance of NMHC2A pTyr^158^ in cytoskeletal organization and cell adhesion, also suggesting its involvement in cell migration.

### 3.4. The Lack of NMHC2A pTyr^158^ Affects Cell Size and Impairs Cell Motility

Considering the well-established roles of Src kinase, NMHC2A, and actin cytoskeleton in the regulation of cell size/shape and migration [76,77,78], and our data showing that the Src-mediated phosphorylation of NMHC2A Tyr^158^ impacts actomyosin cytoskeleton organization, we investigated if the status of NMHC2A pTyr^158^ may affect the cell shape parameters and cell migration. We compared the area and aspect ratio of cells expressing GFP-NMHC2A-WT or -Y158F. Expression of GFP-NMHC2A-Y158F led to a slightly increased cell area (Figure 4A). However, aspect ratio values were similar in both conditions (Figure 4B). To evaluate the consequences of the cytoskeletal alterations induced by the expression of NMHC2A-Y158F, we monitored the motility of HeLa cells using fluorescence timelapse microscopy (Figure 4C and Appendix A). A cell tracking analysis of the timelapse videos revealed that cells expressing GFP-NMHC2A-Y158F migrated with a reduced velocity (Figure 4D) and spanned shorter distances (Figure 4E) when compared to cells expressing GFP-NMHC2A-WT. These results are in line with the elongated focal adhesions and increased stress fibers we observed, indicating that cells are possibly under increased tension if the non-phosphorylatable NMHC2A-Y158F is overexpressed. The NMHC2A pTyr^158^ status thus modulates cell motility, most likely through the control of actomyosin cytoskeleton organization.

### 3.5. NMHC2A pTyr Regulates NM2A Activation and Dynamics

Given our observations, we next examined whether pTyr^158^ could affect the NM2A turnover on actin filaments. To investigate this, we performed Fluorescence Recovery After Photobleaching (FRAP) in HeLa cells overexpressing either GFP-NMHC2A-WT or -Y158F. The GFP signal of NMHC2A was photobleached in a single region per cell where NMHC2A filaments were detected, the recovery of the signal was followed over time and the data were analyzed using the easyFRAP-web on-line platform [37]. The GFP signal within the photobleached region in cells overexpressing GFP-NMHC2A-Y158F recovered faster than that in cells overexpressing GFP-NMHC2A-WT (Figure 5A,B). The mobile fraction, corresponding to the fraction of molecules able to turnover, and the half time required for signal recovery (T-half) were obtained from the easyFRAP-web mean curve fitting tool [37]. The calculated mobile fractions were similar under both conditions (Figure 5C). However, the half time for signal recovery was significantly shorter in cells expressing GFP-NMHC2A-Y158F as compared to those expressing GFP-NMHC2A-WT (Figure 5D), indicating that the expression of non-phosphorylatable NMHC2A-Y158F allows for a faster NM2A turnover.

NM2A filament dynamics and self-organization require NM2A contractility, which is in turn regulated by RLC phosphorylation [2,21]. Thus, we quantified phosphorylated RLC in residues threonine 18 and serine 19 (pMLC) to assess the levels of NM2A activation. In agreement with a faster turnover, we found that cells overexpressing the non-phosphorylatable GFP-NMHC2A-Y158F displayed higher levels of pMLC than those expressing GFP-NMHC2A-WT (Figure 5E–H). In addition, pMLC co-localized with stress fibers, induced by the expression of GFP-NMHC2A-Y158F (Figure 5E). Overall, NM2A-Y158F-expressing cells displayed increased levels of Thr18/Ser19 phosphorylated myosin regulatory light chain and showed a faster turnover of NM2A, which might be associated with an increased contractility [49]. Our results suggest that non-phosphorylated Tyr^158^ triggers NM2A activation and increases its dynamics within assembled filaments and not at the single molecule level.

## 4. Conclusions

Phosphorylations in tyrosine residues control cell signaling and regulate a variety of biological processes, including the reorganization of the cytoskeleton [79,80]. We previously reported an uncharacterized tyrosine phosphorylation event in NMHC2A triggered by bacterial infections [30]. Infection with *L. monocytogenes* induced Src-dependent NMHC2A phosphorylation on Tyr^158^, a residue that is located close to the ATP-binding pocket of NMHC2A and was predicted to be accessible for phosphorylation [81]. Here, we showed that the regulation of NMHC2A pTyr^158^ contributes to the cellular organization of the actomyosin cytoskeleton.

Larger and more elongated focal adhesions, together with increased stress fibers and higher pMLC levels detected in NMHC2A-Y158F-expressing cells, suggest an altered contraction status that may be associated with an increased tension, a stronger adhesion to the substratum, and increased forces exerted on the extracellular matrix [48,82,83]. The phosphorylation status of NMHC2A Tyr^158^ could therefore regulate the assembly/disassembly of focal adhesions and stress fibers, thus providing a novel mechanism to modulate cytoskeletal organization. Phosphorylation events in the regulatory light chain and the heavy chain tail domain, respectively, regulate NM2A activity (reviewed in [2]), the spatial localization [16], and the dynamics of NM2A filaments [84,85]. Whether the phosphorylation of Y158 interferes with other phosphorylation events reported to regulate the NM2A activity or filament assembly is unknown, but worth investigating in the future. To our knowledge, pTyr^158^ is the only phosphorylation occurring in the motor domain of NMHC2A that contributes to regulating the dynamics of the actomyosin cytoskeleton. Previous studies proposed that the formation of heterotypic filaments [86,87] and the assembly of NMII supramolecular structures, named NMII stacks [21,88], may modulate actomyosin dynamics likely to accomplish specific functions in particular spatiotemporal contexts [42,89]. It is possible that pTyr^158^ regulates the assembly of such heterotypic filaments and/or NMII stacks. By interfering with actomyosin remodeling, one can speculate that the phosphorylation status of NMHC2A Tyr^158^ may affect NM2A-driven cellular processes such as cytokinesis, membrane trafficking, and organelle positioning. As a proof of concept, data from our laboratory indicate that permanent phosphorylation of Tyr^158^ impairs cytokinesis in *C. elegans* embryos, while worms expressing a non-phosphorylatable NMHC2A are more susceptible to stress cues such as heat and bacterial infections (Brito et al., unpublished data).

We thus propose a new conserved mechanism for the regulation of the actomyosin cytoskeleton through the control of NM2A pTyr^158^. This adds a new layer of complexity in the control of actomyosin dynamics with a major impact on cell adhesion to a substrate and cell migration.

## Figures and Tables

**Figure 1 cells-12-01871-f001:**
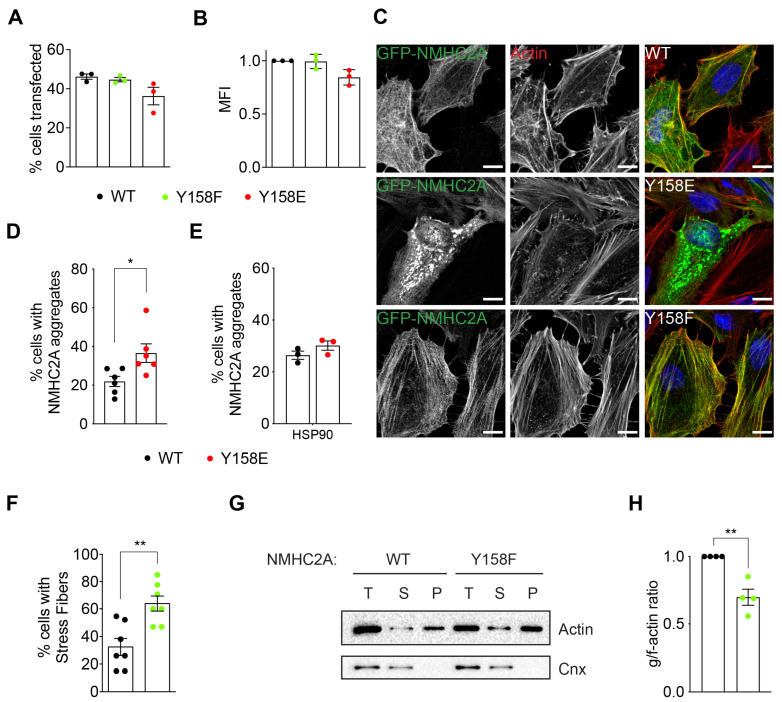
**Actomyosin organization is regulated by the NMHC2A pTyr^158^ status.** (**A**) Percentage of transfected cells determined by the quantification of GFP positive cells by flow cytometry. Each dot corresponds to a single independent experiment. Values are the means ± SEM (n = 3). (**B**) Expression levels of the different variants shown as the MFI (mean fluorescence intensity) of the GFP positive cells determined by flow cytometry. Each dot corresponds to a single independent experiment. Values are the means ± SEM (n = 3). (**C**) Representative confocal microscopy images of HeLa cells ectopically expressing either GFP-NMHC2A-WT, -Y158E, or -Y158F (green) and stained with phalloidin for f-actin (red) and DAPI (blue). Scale bar, 10 µm. (**D**–**F**) Scoring of the percentage of cells displaying (**D**,**E**) NMHC2A aggregation or (**F**) stress fibers in HeLa cells ectopically expressing the indicated NMHC2A variants or co-expressing HSP90 in (**E**). Each dot corresponds to an independent experiment. Values are the means ± SEM; *p*-values were calculated using a two-tailed unpaired Student’s *t*-test; * *p* < 0.05, ** *p* < 0.01. (**G**) Immunoblots showing the levels of globular (g)- and filamentous (f)-actin in HeLa cells ectopically expressing NMHC2A WT or Y158F variants. g- and f-actin from the total cell lysates (T) were separated by ultracentrifugation. g-actin was recovered from supernatant fractions (S) while f-actin remained associated with pellet fractions (P). Calnexin was used as a loading control (Cnx). (**H**) The g-/f-actin ratio quantified from immunoblot images. Each dot corresponds to an independent experiment. Values are the means ± SEM (n = 4); the *p*-value was calculated using a two-tailed unpaired Student’s *t*-test; ** *p* < 0.01.

**Figure 2 cells-12-01871-f002:**
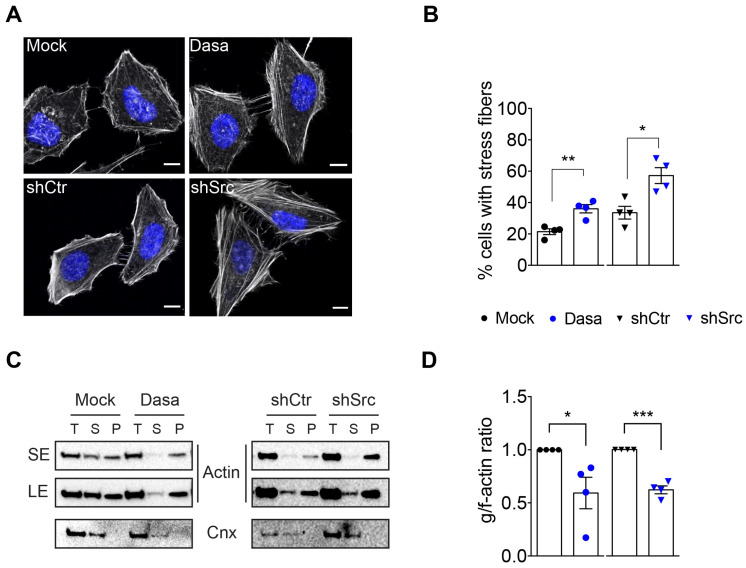
**Src kinase controls actomyosin cytoskeleton organization in HeLa cells.** (**A**) Confocal microscopy images of control (mock and shCtr) and Src-impaired (Dasatinib-treated and shSrc) HeLa cells stained with phalloidin for f-actin (greyscale) and with DAPI (blue). Scale bar, 10 µm. (**B**) Percentage of cells with stress fibers quantified from images similar to those shown in (**A**). Values are the means ± SEM (n = 4); *p*-values were calculated using a two-tailed unpaired Student’s *t*-test, * *p* < 0.05, ** *p* < 0.01. (**C**) Immunoblots showing the levels of globular (g)- and filamentous (f)-actin in HeLa cells treated as in (**A**). g- and f-actin from the total cell lysates (T) were separated by ultracentrifugation. g-actin was recovered from supernatant fractions (S) while f-actin was associated with pellet fractions (P). Calnexin was used as a loading control (Cnx). Actin levels are shown with both short exposure (SE) and long exposure (LE). (**D**) Quantification of the g-/f-actin ratio was conducted using immunoblot signals. Values are the means ± SEM (n = 4); *p*-values were calculated using a two-tailed unpaired Student’s *t*-test, * *p* < 0.05, *** *p* < 0.001. In (**B**,**D**), each dot or triangle corresponds to an independent experiment.

**Figure 3 cells-12-01871-f003:**
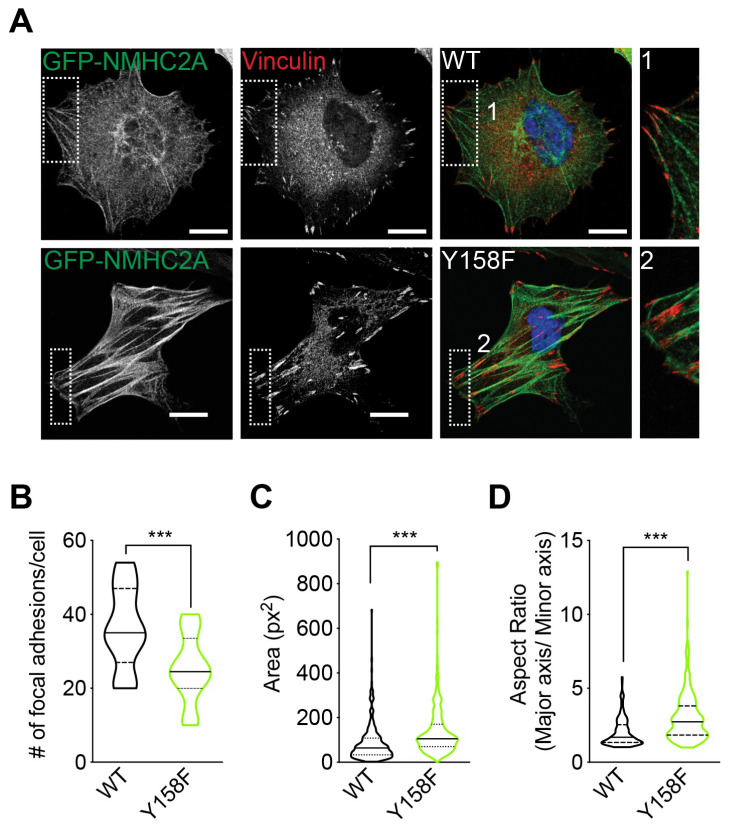
**The number and the morphology of focal adhesions in HeLa cells are determined by the phosphorylation status of NMHC2A in Tyr^158^.** (**A**) Representative confocal microscopy images of HeLa cells ectopically expressing GFP-NMHC2A-WT or GFP-NMHC2A-Y158F (green), immunolabeled for vinculin (red) and stained with DAPI (blue). The insets (1 and 2) show the focal adhesion morphology. Scale bar, 10 µm. (**B**) Quantification of the number of focal adhesions per cell from images similar to those shown in (**A**). Each dot corresponds to the mean number of focal adhesions per cell obtained from three independent experiments. Values are the means ± SEM; *p*-value was calculated using a two-tailed unpaired Student’s *t*-test, *** *p* < 0.001 (**C**,**D**) Quantification of the focal adhesion (**C**) area and (**D**) aspect ratio from images similar to those shown in A. Each dot corresponds to a single focal adhesion. Values are the means ± SEM (n > 140); *p*-values were calculated using a two-tailed unpaired Student’s *t*-test, *** *p* < 0.001.

**Figure 4 cells-12-01871-f004:**
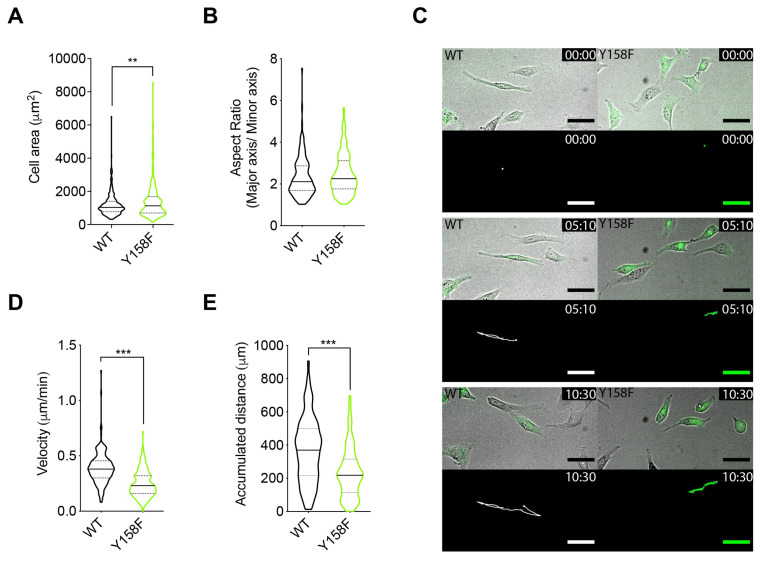
**The lack of NMHC2A pTyr^158^ impairs the motility of HeLa cells.** (**A**,**B**) Quantification of cell shape parameters of HeLa cells ectopically expressing GFP-NMHC2A-WT or -Y158F: (**A**) area and (**B**) aspect ratio. Cells were stained with phalloidin and quantifications were performed using Fiji™. Each dot represents a single cell. Values are the means ± SEM (n > 160); *p*-values were calculated using a two-tailed unpaired Student’s *t*-test, ** *p* < 0.01. (**C**) Sequential frames of the timelapse microscopy Appendix A, from 0 to 10 h 30 min. HeLa cells ectopically expressing GFP-NMHC2A-WT or GFP-NMHC2A-Y158F are shown. The movement of individual cells over time was analyzed using the Fiji™ Manual Tracking plugin and is represented by colored lines. Scale bar, 50 μm. (**D**,**E**) Quantification of cell motility parameters using the Manual Tracking plugin of Fiji™: (**D**) velocity and (**E**) accumulated distance in HeLa cells as in A. The data were obtained from movies similar to Appendix A. Each dot represents a single cell. Values are the means ± SEM (n > 200); *p*-value was calculated using a two-tailed unpaired Student’s *t*-test, *** *p* < 0.001.

**Figure 5 cells-12-01871-f005:**
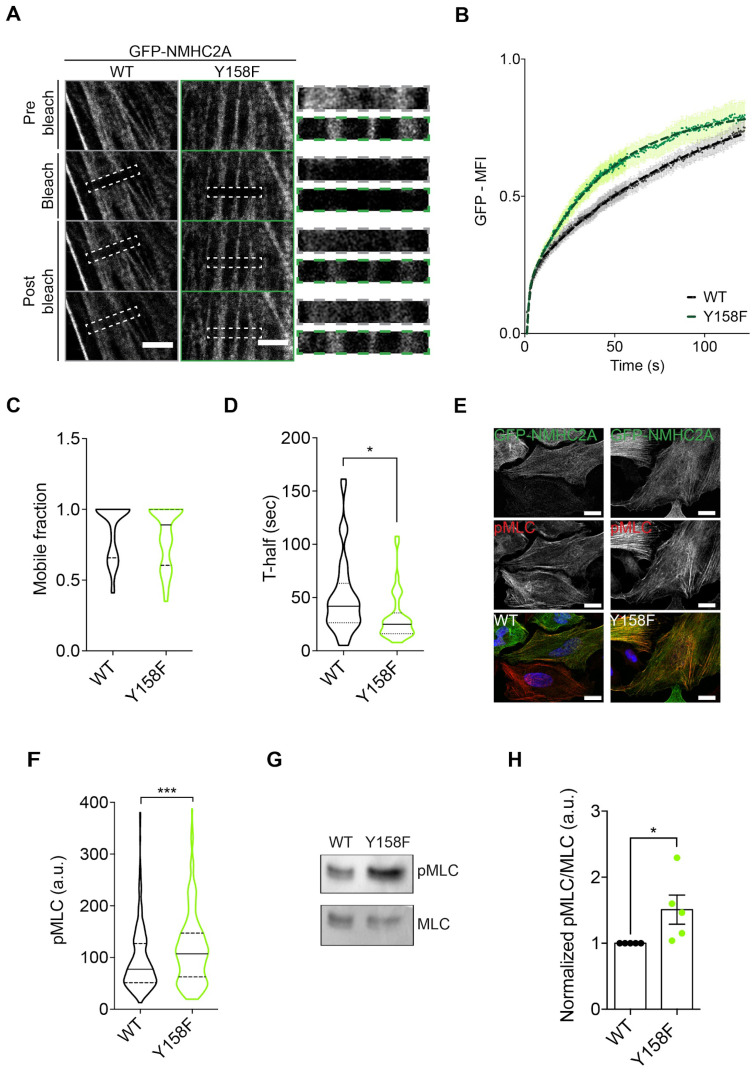
**NMHC2A pTyr^158^ controls the dynamic assembly of bipolar filaments in HeLa cells.** (**A**) Representative confocal microscopy timelapse images of Fluorescence Recovery After Photobleaching (FRAP) experiments in HeLa cells ectopically overexpressing either GFP-NMHC2A-WT or -Y158F. Insets of the bleached region in dashed lines. Scale bar, 5 μm. (**B**) Plot of the normalized mean fluorescence intensity recovery curves obtained for each condition tested (black, WT; green, Y158F). Values are the means ± SEM (n ≥ 40 cells). (**C**,**D**) Quantification of the (**C**) mobile fraction and the (**D**) half-maximal recovery time (T-half) obtained from the curve fitting of each individual recovery curve. Each dot represents a single cell. Values are the means ± SEM (n ≥ 40 cells); *p*-values were calculated using a two-tailed unpaired Student’s *t*-test, * *p* < 0.05 (**E**) Representative confocal microscopy images of HeLa cells ectopically expressing either GFP-NMHC2A-WT or -Y158F (green), immunolabeled to detect pMLC (red) and stained with DAPI (blue). Scale bar, 15 µm. (**F**) Levels of pMLC in cells expressing the different NMHC2A variants, quantified using Fiji™. Each dot corresponds to a single cell. Values are the means ± SEM (n > 300); *p*-value was calculated using a two-tailed unpaired Student’s *t*-test, *** *p* < 0.001. (**G**) Immunoblot showing the levels of phosphorylated MLC (pMLC) and the total levels of MLC in whole cell lysates of HeLa cells expressing either GFP-NMHC2A-WT or -Y158F. (**H**) Quantification of pMLC immunoblot signals normalized to that of MLC. Each dot corresponds to an independent experiment. Data correspond to means ± SEM (n = 5); *p*-value was calculated using a two-tailed unpaired Student’s *t*-test, * *p* < 0.05.

## Data Availability

Not applicable.

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
