# Peer review of "Src-Dependent NM2A Tyrosine Phosphorylation Regulates Actomyosin Remodeling"

_cells, 2023, doi:10.3390/cells12141871_

Round 1

Reviewer 1 Report

In this manuscript Brito et al., have reported the role of tyrosine158 phosphorylation in NMIIA head domain in regulating F-actin formation, focal adhesion formation, cell migration and kinetics of bipolar filament accumulation on F-actin filaments, using Hela cells as a model system. While the experiments are well designed and have appropriate controls, there are some concerns that the authors need to address to ensure that the manuscript is ready for publication.

Major concerns:

1.     In figure 1, it has been shown that lack of phosphorylation of NMIIA at Tyr158, significantly enhances F-actin filaments. Does this also affect how cells exert force on the extracellular matrix?

2.     In figure 2, Src kinase knock down shows enhanced F-actin formation. The authors need to show immunoblots to confirm the efficacy of Src kinase known with shRNA.

3.     Furthermore, the authors also need to confirm that Src kinase knock down leads to lack to phosphorylation at Tyr158 in NMIIA.

4.     How does phosphorylation of NMIIA at Tyr158 or lack of phosphorylation of NMIIA at Tyr158 affect the dimensions of NMIIA bipolar filaments?

Quality of english language use in the manuscript is appropriate 

Author Response

Reviewer 1

In this manuscript Brito et al., have reported the role of tyrosine158 phosphorylation in NMIIA head domain in regulating F-actin formation, focal adhesion formation, cell migration and kinetics of bipolar filament accumulation on F-actin filaments, using Hela cells as a model system. While the experiments are well designed and have appropriate controls, there are some concerns that the authors need to address to ensure that the manuscript is ready for publication.

We are thankful to the reviewer for her/his time and interest in our manuscript and for the raised issues, which we believe allowed us to improve the quality and the impact of our manuscript. We provide below a point-by-point response to the raised questions and modified the manuscript according to the reviewer's suggestions. Text modifications are shown with track changes in the revised version of our manuscript.

Major concerns:

  1. In figure 1, it has been shown that lack of phosphorylation of NMIIA at Tyr158, significantly enhances F-actin filaments. Does this also affect how cells exert force on the extracellular matrix?

The reviewer raises an interesting issue. Current data indicate that MRLC phosphorylation leads to the activation of NM2A motor activity and its assembly into bipolar filaments associated with f-actin to ultimately generate contraction (Sao et al., 2019). The tension generated was previously shown to result in the maturation and enlargement of focal adhesions (FAs) (Pasapera et al., 2010), which link cells to the extracellular matrix (ECM) and transmit mechanical forces between the actin cytoskeleton and ECM.  Other studies propose that increased focal adhesions size results in increased tension across integrins to the ECM (Doyle et al., 2022, Chien et al., 2022).  As a consequence, we speculate that, since the expression of the NMHC2A Y158F variant results in increased FAs area and aspect ratio, and increased stress fibers, cells overexpressing such mutant might exert increased force on the ECM.

To properly assess this issue we would need to perform Traction Force Microscopy. Unfortunately, we lack this technology in our institute and even in our country. Thus we won't be able to perform the experiments in a timely manner. Alternatively, we could perform Atomic Force Microscopy (AFM), but this would only provide information on the mechanical properties of the cells and not specifically determine the forces exerted on the ECM, and also would require optimizations and time.

In conclusion, we acknowledge the issue raised by Referee 1 and agree on its relevance. We now mention this issue in the discussion of the results (line 475).

  1. In figure 2, Src kinase knock down shows enhanced F-actin formation. The authors need to show immunoblots to confirm the efficacy of Src kinase knockdown with shRNA.

We agree that the efficacy of Src kinase knock down needs to be shown. The cells knocked down for the expression of Src kinase (HeLa shSrc) were used in different published and unpublished studies by our lab (Almeida et al., 2015, Fig 1). We always control their deficiency in Src expression by immunoblot. We provide, as Fig S2, a blot performed in parallel with experiments shown in Fig 2, showing that shSrc cells only express residual levels of Src kinase. This information was added in the revised version of our manuscript (lines 286-287).

  1. Furthermore, the authors also need to confirm that Src kinase knock down leads to lack of phosphorylation at Tyr158 in NMIIA.

We acknowledge the point raised by the reviewer. However, the phosphorylation of Tyr158 in NM2A is barely detected in non-stimulated cells. Thus, demonstrating that pTyr158 is lacking in cells knocked down for the expression of Src kinase (shSrc) might not be possible under steady state conditions. However, we previously reported that phosphorylation of Tyr158 in NM2A is promoted by the infection of epithelial cells by bacterial pathogens (Almeida et al., 2015, Figs 1 and 2) and we demonstrated that such phosphorylation does not occur in cells that express reduced levels of Src kinase. In addition, we recently found that pore-forming toxins may also trigger phosphorylation of Tyr158 in NM2A (unpublished), which does not occur in cells expressing reduced levels of Src kinase (Brito et al., 2021 -biorxiv, Figure 3). A sentence indicating that we had previously shown that the lack of Src kinase impairs phosphorylation of Tyr158 was added in the revised version of the manuscript (lines 296-298).

  1. How does phosphorylation of NMIIA at Tyr158 or lack of phosphorylation of NMIIA at Tyr158 affect the dimensions of NMIIA bipolar filaments?

This is a very interesting question . Considering that our data suggest that non-phosphorylated Tyr158 triggers NM2A activation and increases its dynamics within assembled filaments, one may postulate that bipolar filaments might display different lengths dependent on the phosphorylation status of Y158. NM2A filaments can be measured by super-resolution microscopy using a double-labeling strategy to label simultaneously the head and the tail of NM2A molecules with two different fluorophores (Weibenbruch et al., 2022, Fenix et al., 2016, Jiu et al., 2019). In our experimental set up we would first need to transfect the cells with the GFP-NMHC2A-WT or Y158F variants, which are not suitable to use under STED microscopy (the one available in our institute). Thus, unfortunately, we are not able to perform these experiments in a timely manner. Nevertheless, we acknowledge the suggestion and do believe that this experiment would fit a story in preparation where we look at myosin activity in vivo and can correlate the in vitro activity observations with the assembly of bipolar filaments and stacks of NM2A in wildtype and Y158F backgrounds.

Reviewer 2 Report

The manuscript by Brito et al. investigates the role of NM2A tyrosine phosphorylation at residue 158 (pTyr158) in the regulation of actomyosin cytoskeleton organization, cell adhesion and migration. The authors show that pTyr158 is induced by Listeria monocytogenes infection and depends on Src kinase activity. They also demonstrate that the expression of a non-phosphorylatable NM2A mutant (Y158F) leads to increased stress fibers, larger focal adhesions, reduced cell motility and higher NM2A activity. Furthermore, they perform FRAP experiments to show that pTyr158 affects NM2A dynamics and turnover. The authors propose that Src-dependent NM2A pTyr158 is a novel layer of regulation of the actomyosin cytoskeleton organization.

The manuscript is well-written, clear and concise. The results are supported by appropriate methods and analyses. The figures are of good quality and informative. The discussion is comprehensive and provides relevant insights into the molecular mechanisms and physiological implications of NM2A phosphorylation. The authors also acknowledge the limitations and challenges of their study and suggest future directions.

The manuscript presents novel and significant findings that advance our understanding of NM2A regulation and function in cytoskeletal remodeling and cellular processes. The study is original, rigorous and timely. I recommend the manuscript for publication in Cells after minor revisions.

Some minor points that could be addressed are:

  1. In the discussion, it would be relevant to discuss how NM2A pTyr158 may interact with other known phosphorylation events that regulate NM2A activity and filament formation.

  2. In the discussion, it would be important to mention how NM2A pTyr158 may affect other cellular processes that depend on actomyosin remodeling, such as cytokinesis, endocytosis, exocytosis, membrane trafficking and organelle positioning.

Author Response

Reviewer 2

The manuscript by Brito et al. investigates the role of NM2A tyrosine phosphorylation at residue 158 (pTyr158) in the regulation of actomyosin cytoskeleton organization, cell adhesion and migration. The authors show that pTyr158 is induced by Listeria monocytogenes infection and depends on Src kinase activity. They also demonstrate that the expression of a non-phosphorylatable NM2A mutant (Y158F) leads to increased stress fibers, larger focal adhesions, reduced cell motility and higher NM2A activity. Furthermore, they perform FRAP experiments to show that pTyr158 affects NM2A dynamics and turnover. The authors propose that Src-dependent NM2A pTyr158 is a novel layer of regulation of the actomyosin cytoskeleton organization.

The manuscript is well-written, clear and concise. The results are supported by appropriate methods and analyses. The figures are of good quality and informative. The discussion is comprehensive and provides relevant insights into the molecular mechanisms and physiological implications of NM2A phosphorylation. The authors also acknowledge the limitations and challenges of their study and suggest future directions.

The manuscript presents novel and significant findings that advance our understanding of NM2A regulation and function in cytoskeletal remodeling and cellular processes. The study is original, rigorous and timely. I recommend the manuscript for publication in Cells after minor revisions.

We are thankful to the reviewer for her/his time and interest in our manuscript and for her/his suggestions. We are also glad about her/his positive comments on our work. Our revised version now includes in the discussion the reviewer's suggestions. Text modifications are shown with track changes in the revised version of our manuscript.

Some minor points that could be addressed are:

  1. In the discussion, it would be relevant to discuss how NM2A pTyr158 may interact with other known phosphorylation events that regulate NM2A activity and filament formation.

This is an interesting point. The activity, the localization and the polymerization of NM2A have been reported to rely on phosphorylation/dephosphorylation events (for review Brito and Sousa, 2020). Phosphorylations on the Regulatory Light Chain are amongst the best characterized:

  1. phosphorylation on Ser 19 and Thr 18 were associated with increased activity and filament formation;
  2. phosphorylation on Ser 1, Ser 2 and Thr 9 are inhibitory as they were associated with a decreased rate of ATP hydrolysis in vitro, and stress fiber disassembly and mitotic arrest in cells. These phosphorylations allosterically reduce Ser19 phosphorylation, thus decreasing the ATPase activity;
  3. phosphorylation of Tyr155 was recently demonstrated to spatially control the assembly and function of NM2 in migrating cells. This phosphorylation impairs the interaction between NMHC2s and RLCs, specifically at migratory cell protrusions, restricting the assembly of functional NMII molecules.

The assembly of NM2A filaments assembly is also regulated by phosphorylation events on the coiled-coil tail and non-helical tailpiece, specifically on Ser 1916 and Ser 1943.

In our study, we only assessed the crosstalk between the phosphorylation of Tyr158 and Ser 19/Thr 18. We found that cells overexpressing the non-phosphorylatable version Y158F, display higher levels of phosphorylated Ser 19/Thr 18 (pMLC, Figure 5), which points to increased NM2A activation and dynamics within filaments. Whether the phosphorylation of Tyr 158 may interfere with other reported phosphorylation events was not evaluated, but we agree that this could be worth doing in the future.

This is now mentioned in the revised version of the manuscript (lines 471-473 and 481-483).

  1. In the discussion, it would be important to mention how NM2A pTyr158 may affect other cellular processes that depend on actomyosin remodeling, such as cytokinesis, endocytosis, exocytosis, membrane trafficking and organelle positioning.

We fully agree with the reviewer. We indeed previously formulated the hypothesis that the phosphorylation status of Tyr158 could affect cellular processes driven by NM2A. We have data indicating that the status of phosphorylation in residue Tyr158 may affect NM2A cellular functions. This is now discussed in the new version of the manuscript (lines 489-495).

Round 2

Reviewer 1 Report

While the changes to the manuscript are definitely welcome, any data showing phosphorylation at Try158 residue of NMIIA would make it better.